# A Cost-Benefit Analysis of the COVID-19 Asymptomatic Mass Testing Strategy in the North Metropolitan Area of Barcelona

**DOI:** 10.3390/ijerph18137028

**Published:** 2021-06-30

**Authors:** Francesc López Seguí, Oriol Estrada Cuxart, Oriol Mitjà i Villar, Guillem Hernández Guillamet, Núria Prat Gil, Josep Maria Bonet, Mar Isnard Blanchar, Nemesio Moreno Millan, Ignacio Blanco, Marc Vilar Capella, Martí Català Sabaté, Anna Aran Solé, Josep Maria Argimon Pallàs, Bonaventura Clotet, Jordi Ara del Rey

**Affiliations:** 1Directorate for Innovation and Interdisciplinary Cooperation, North Metropolitan Territorial Authority Catalan Institute of Health, 08916 Badalona, Spain; oestrada@gencat.cat; 2Fight AIDS and Infectious Diseases Foundation, 08916 Badalona, Spain; omitja@flsida.org; 3Centre de Recerca en Economía de la Salut, Pompeu Fabra University, 08005 Barcelona, Spain; guillemhg98@gmail.com; 4North Metropolitan Primary Care Directorate, Catalan Institute of Health, 08916 Badalona, Spain; nprat@gencat.cat (N.P.G.); jmbonets@gencat.cat (J.M.B.); misnard.bnm.ics@gencat.cat (M.I.B.); nmorenom.bnm.ics@gencat.cat (N.M.M.); iblanco.germanstrias@gencat.cat (I.B.); mvilar@gencat.cat (M.V.C.); aaran@catsalut.cat (A.A.S.); gterritorial.mn.ics@gencat.cat (J.A.d.R.); 5Comparative Medicine and Bioimage Centre of Catalonia (CMCiB), Fundació Institut d’Investigació en Ciències de la Salut Germans Trias i Pujol, 08916 Badalona, Spain; mcatala@igtp.cat; 6Public Health Secretary, Ministry of Health of Catalonia, 08007 Barcelona, Spain; jmargimon@gencat.cat; 7IrsiCaixa—Institut de Recerca de La SIDA, Hospital Universitari Germans Trias I Pujol, 08916 Badalona, Spain; bclotet@irsicaixa.cat

**Keywords:** test-tracking-quarantine, cost benefit analysis, economic analysis, COVID-19, asymptomatic screening, mass testing, non-pharmacological interventions

## Abstract

Background: The epidemiological situation generated by COVID-19 has highlighted the importance of applying non-pharmacological measures in the management of the epidemic. Mass screening of the asymptomatic general population has been established as a priority strategy by carrying out diagnostic tests to detect possible cases, isolate contacts, cut transmission chains and thus limit the spread of the virus. Objective: To evaluate the economic impact of mass COVID-19 screenings of an asymptomatic population during the first and second wave of the epidemic in Catalonia, Spain. Methodology: Cost-Benefit Analysis based on the estimated total costs of mass screening versus health gains and associated health costs avoided. Results: Excluding the value of monetized health, the Benefit-Cost ratio was estimated at 0.45, a low value that would seem to advise against mass screening policies. However, if monetized health is included, the ratio is close to 1.20, reversing the interpretation. In other words, the monetization of health is the critical element that tips the scales in favour of the desirability of screening. Results show that the interventions with the highest return are those that maximize the percentage of positives detected. Conclusion: Efficient management of resources for the policy of mass screening in asymptomatic populations can generate high social returns. The positivity rate critically determines its desirability. Likewise, precocity in the detection of cases will cut more transmissions in the chain of contagion and increase the economic return of these interventions. Maximizing the value of resources depends on screening strategies being accompanied by contact-tracing and specific in their focus, targeting, for example, high-risk subpopulations with the highest rate of expected positives.

## 1. Introduction

Early detection of cases with SARS-Cov-2 infection is a key factor in controlling the transmission of the disease at the community level. Therefore, since the start of the current pandemic, health systems around the world have devoted numerous resources to various types of mass screening strategies [1,2,3,4,5,6,7]. In the year since March 2020, almost 360 million tests have been carried out in Europe for the detection of the virus, amounting to about 6.5 million tests per week: during the same period Catalonia, with a population of 7.6 million, has performed almost 5 million. In most contexts, the identification of infected people through these tests is complemented by subsequent contact-tracing and quarantine isolation -what is known as TTQ (Test-Tracking-Quarantine) strategies.

Infected persons without symptoms -whether presymptomatic or asymptomatic- may account for roughly 40% of all transmission [8,9,10]. During the first months of the pandemic, however, the material and personnel required to carry out testing of the asymptomatic general population was not available. The increased availability of resources and the worsening of the epidemiological situation during the summer and early autumn of 2020, with many of the new cases asymptomatic, led the Department of Health of the Catalan autonomous government to establish as a priority strategy mass screening using Polymerase Chain Reaction (PCR) tests and/or Rapid Antigen Tests (RAT) with the aim of detecting as many cases as possible, cutting transmission chains, isolating contacts and thus limiting the spread of the virus. The policy was also motivated by the previous mass screenings carried out in countries such as Korea, Taiwan, Japan, China, New Zealand and the Czech Republic, which proved successful in controlling the epidemic [6,11,12,13,14,15,16,17,18]. On the one hand, the lower prevalence of infection in asymptomatic testing implies that more resources need to be devoted to identifying a positive case and cutting the corresponding potential transmission chain potential. On the other hand, detecting positive cases not suspected of being infected can help prevent the progression of the virus in a population more likely to take risks.

While it is true that some of these mass screenings have been criticized because of the shortage of evidence in their support, their lack of planning and the great expense they have involved [4,19,20,21], several economic analyses have shown that their costs are largely offset by their benefits. For example, studying mass screenings that were implemented in the US, Cutler and Summers estimated that their economic benefits were about 30 times their cost [22]. In another study, Atkeson et al. estimated the benefit-cost ratio to be in the range of 2 to 15 [23]. Adopting an approach much more similar to the conventional framework used to measure the economic impact of health interventions [24] and applying it to the case of mass testing in Spain, González López-Valcárcel and Vallejo-Torres calculated an approximate cost-benefit ratio of 7 to 19 [25]. There is no evidence, however, regarding the economic impact of mass screening in asymptomatic populations. In this context, the aim of this study is to undertake a cost-benefit analysis of the mass COVID-19 screenings in the asymptomatic general population carried out in Catalonia during the first and second wave of the current epidemic.

## 2. Methodology

### 2.1. Setting

The object of the study is the analysis of the series of mass screenings of the general asymptomatic population carried out under the auspices of the Catalan Institute of Health’s General Directorate in the North Metropolitan health zone, the most heavily populated district of the greater Barcelona metropolitan area, with a total of 1,986,032 inhabitants, representing 25.9% of the total population of Catalonia, as documented by the General Directorate of Public Health of Catalonia. Of the various so-called health zones into which Catalonia is divided, this is the one where the largest number of mass COVID-19 screenings have taken place. In fact, some of the screening has specifically targeted health professionals, staff and residents in care homes for the elderly but these screenings have not been included in the present study because of the very narrow focus of the tested population, as opposed to the testing of the general public of interest here.

The sample under analysis here thus consisted of 78 mass screenings carried out up to the end of 2020, corresponding to 125,865 individual tests, 1719 of which were positive for COVID-19 infection, representing 1.37% of the total. Screenings took place in four types of context, school (12/78), community (58/78), school and community (6/78) and workplace (2/78), and were performed either during the first wave de-escalation (July–August 2020, 20/78) or the second wave (September–December 2020, 58/78). The assays employed either PCR test (67/78) or RAT (11/78). Table 1 shows the number of tests performed and positives detected for each of the methods used.

### 2.2. Study Parameters

Taking the López-Valcárcel and Vallejo-Torres model [25] as a reference, the effectiveness of the screening strategy was measured by estimating the number of cases avoided, which depends on the set of parameters shown in Table 2. Of these, the parameters that were determined empirically were: the cost, number and rate of positive tests; the effective reproductive number (R_t_, expected number of new infections caused by an infectious individual in a population where some individuals may no longer be susceptible) at the time of screening (using the average of all days when screening was underway); the number of contact-tracers employed (the part proportionate to the population included in the territory) and their daily cost; and the costs of hospitalization and admission to the ICU (paid for by the healthcare service contractor). Regarding the use of resources, the figures used here for hospitalization, ICU occupancy and mortality rates declared for the Catalan territory were taken after 11 May 2020 and therefore exclude the significantly higher rates observed during the first wave of the epidemic in Spain. With regard to the transmission potential at the time of detection (number of iterations of infections), it was calculated based on the average position of the screening date in the epidemic curve (in terms of the cumulative incidence at 14 days) in a range between 0 and 5 (relative to the approximate duration of a wave and assuming that a contagion cycle lasts 15 days). The value attributed to the set of screenings was calculated with a weighted average according to the number of tests performed.

Values were assumed rather than observed for the remaining parameters, namely: the number and rate of close contacts testing positive; the cost of follow-up for COVID-19 cases that do not require hospitalization; the quarantine adherence rate; and the proportion of people detected who could become infected after being detected (all based on [25]). Following the reference model, we assume that one-third of COVID-19 cases requiring hospitalization will experience some type of long-term health complication; the healthcare costs associated with these consequences were calculated assuming an annual incremental cost of €1000 for the remaining life expectancy of patients experiencing long-term complications, discounted at a rate of 3%. Finally, using the same parameters as in the aforementioned study, gains were measured in Quality-Adjusted Life Years (QALYs) associated with deaths and cases of long-term morbidity avoided as a result of the mass screening, and monetized according to the reference study [25]. Productivity costs related to long-term mortality and morbidity were not considered because the average age of infected citizens who died or had moderate or severe symptoms is close to or higher than the official retirement age in Spain [26,27].

## 3. Results

Table 3 shows the results for the baseline scenario (all screenings). It will be seen that the total cost of this series of screenings was €8,372,265, of which 87.67% corresponds to costs directly associated with the tests, while the remainder consists of costs derived from the employment of contact tracers. Collectively, the mass screenings analysed here identified a total of 1724 positives, which according to the model would imply an estimated total of 5429 additional infections prevented, which in turn would imply the avoidance of 168 hospitalizations, 11 ICU admissions, 56 cases with permanent sequelae, and 33 deaths, the remaining 5161 being cases that could be treated at home. These cases avoided represent a saving of €3,762,322 in the use of health resources, with 38%, 27%, 13% and 22% of this value corresponding to COVID-19 cases treated at home, hospitalizations, ICUs occupancies and cases with permanent sequelae, respectively. Thus, without monetizing the QALY, the intervention represents a deficit of €4,609,943. If we do monetize the 156 and 95 QALY corresponding respectively to the morbidity and mortality avoided, we must add to the benefits €6,266,368 in terms of improvements in the state of health of the population (more than two thirds of this figure corresponding to the avoided morbidity). In this case, the benefits of the intervention (€10,028,690) would exceed the aforementioned costs (€8,372,265).

These figures imply two main results. On the one hand, the benefit-cost ratio that excludes the monetized health of a mass screening intervention in the asymptomatic general population approaches 0.45. In other words, the social return on the investment of one euro is 45 cents. From this perspective, the analysis would seem to advise against carrying out mass screening. In contrast, if monetized health is included in the calculation, the benefit-cost ratio approaches 1.20, reversing the interpretation. Thus, incorporating the monetization of health is a critical element that, from the perspective of this analysis, tips the scales in favour of the desirability of mass screening. This logic applies equally to the rest of the scenarios analysed. In general, higher positivity rates, lower cost of tests, higher R_t_ and the higher number of potential iterations by which the number of avoided cases is calculated (initial stages of the curve) are related to a greater effectiveness of the screenings policy.

### Sensitivity Analysis

The same analysis methodology was applied in four other scenarios (Table 4). With regard to the screenings performed with the PCR and RAT tests, it should be borne in mind that the difference in the results, relative to the base case, of these scenarios is attributable not only to the type of test but also to the set of conditions in which these tests were used (e.g., RAT tests were available only after the start of the second wave). Thus, we find that the results are slightly better for RAT than PCR tests: the lower the cost and the higher the rate of positive close contacts, the more they compensate for an R_t_ and a lower positivity rate. With respect to the scenarios that differentiate the first and second waves, the results show that the lower positivity and R_t_ rates during the second wave imply worse results in the benefit-cost ratios analysed. Generally speaking, the same trend applies in all scenarios. When analysed separately, screenings performed in school, community, school + community and workplace contexts did not show significantly different results.

## 4. Discussion

Since February 2020, SARS-Cov-2 has been communally transmitted in most industrialized countries. Given the practical and legal obstacles to imposing sufficiently strict and long-term community-wide confinements that could permanently stop transmission, most countries have chosen to combine short and long-term confinements in periods when communal transmission is increasing or already high [1,2,3]. In such contexts, community screening has been applied as a complement to confinement to try to reduce transmission rates. Ideally, from the point of view of their effectiveness, such mass screening needs to be massive, systematic and performed early in the epidemic wave to try to reach as much of the population as possible; contact-tracers must be employed to locate and alert the close contacts of positives detected; and the total isolation of positives and quarantining of contacts must be ensured. From the point of view of efficiency, in a context of scarcity of resources (often the case during the first wave), it is necessary to analyse which interventions have maximal impacts relative to their costs. Our analysis has suggested that the incorporation of the monetization of health impacts is the critical element that, from an economic point of view, tips the scales in favour of its desirability. As reported in previous studies, the inclusion of these impacts triples the social return of COVID-19 mass screening.

Unlike these previous studies, the testing positivity rate we report here (1.37%) is significantly lower than that reported elsewhere (5% assumed in Arkenson et al. and Cutler & Summers and 10% assumed in González López-Valcárcel & Vallejo-Torres [22,23,25]). This is because the screening policy analysed here was performed on the asymptomatic general population. Previous studies have been based on hypothetical scenarios where various groups for testing (e.g., asymptomatic, symptomatic, contacts) are mixed. In this sense our analysis, which uses empirical values for the specific case of an asymptomatic population testing, is arguably more robust.

However, taken together, the results from both the present study and the aforementioned research indicate that there is room for improvement in maximizing the value of resources devoted to the policy of mass screening in asymptomatic populations. The more specific the measures for detecting companies, the more social return they will have: mass screening of the entire population does not seem the best tactic, in economic terms, even if it has a return greater than 1. The best strategy must be to detect at minimal cost the maximum number of asymptomatic cases early (i.e., a proactive and localized screening model). Screening asymptomatic individuals indiscriminately involves a very high opportunity cost whereas doing so in a highly focused manner will be in comparison much more efficient.

Whatever the economic return, screening actions have intangible impacts. They can have an educational and reassuring social effect, promote social awareness and other non-pharmacological self-protection measures (e.g., handwashing, social distancing, mask use, adequate ventilation) and increase the perception of coordinated government action in the face of the epidemic [21,28]. But there is a danger that mass screening may reflect more a political impulse to show the public that the health institutions are “doing something” about the epidemic than a policy based on firm scientific evidence. Furthermore, the effectiveness of screening may be conditioned by a gap between design and execution. The mix of inputs needed to organize these policies requires a changing availability of resources: in mass community screening, PCR testing implies a laboratory response time not longer than 24 h, while RAT testing involves more on-site staff to analyse and record tests performed in the clinical histories of all individuals screened. Assuming these reasonable limitations, to the extent possible, preventive policies related to COVID-19 testing should be modelled in accordance with their proven cost-effectiveness. In general terms, as a guide for making decisions about future screening strategies, the higher the positivity rate, R_t_ and number of iterations for which the scenario is calculated, the greater the effectiveness, in terms of economic return, of the screening policy.

### Limitations

In this analysis, the costs of the PCR and TAR tests were calculated based on what the healthcare contractor actually paid in this case. Given that these prices were tentatively set, they may not reflect the true cost to the provider doing the screening, and might actually be much less, perhaps even 30% of the figures used here. If this were the case, the analysis would yield results approximately 50% more favourable than what we have calculated here. Instead, the model does include the time cost of individuals who took the test. If added, these would decrease the benefit-cost ratio.

In relation to the epidemiological model used, several factors must be taken into account. First, it does not consider the variability of various scenarios or the interaction with other non-pharmacological measures that can attenuate the theoretical spread rate expected in an epidemic outbreak. Secondly, several studies point to the possibility that asymptomatic patients have a lower contagion capacity than symptomatic ones [29,30], so we could be overestimating the potential number of infections because the first transmission chain would be smaller. Finally, it should be mentioned that there are discrepancies in data between [26,27] with regard to the estimation of hospitalization, ICU occupancy and mortality rates. In this study we used the lower of the two estimates [27], which may have reduced the benefit-cost ratio we report.

## 5. Conclusions

COVID-19 screening programs need to be modelled on successful testing interventions. Mass testing is one of the non-pharmacological strategies that has been shown to be most effective in managing the epidemic. The results of this study show that an efficient implementation of mass screening in asymptomatic populations can generate high social returns, with the inclusion of avoided impacts on health being the factor that critically determines its desirability. Maximizing the value of dedicated resources depends on screening strategies being accompanied by contact-tracing and specific in their focus, targeting, for example, high-risk subpopulations with the highest rate of expected positives.

## Figures and Tables

**Table 1 ijerph-18-07028-t001:** Screenings in asymptomatic population. Northern Metropolitan Health Region. July–December 2020.

	PCR Tests	PCR Positives	% Positives	RAT Tests	RAT Positives	% Positives	Total Tests	Total Positives	Total % Positives
1st wave	27,570	535	1.94%				27,570	535	1.94%
2nd wave	66,435	1021	2.26%	31,860	163	0.51%	98,295	1184	1.20%
Community	53,472	749	1.40%	31,860	163	0.51%	85,332	912	1.07%
Community + school	2484	41	1.65%				2484	41	1.65%
School	10,075	213	2.11%				10,075	213	2.11%
Workplace	404	18	4.46%				404	18	4.46%
Overall total	94,005	1569	1.40%	31,860	163	0.51%	125,865	1719	1.37%

**Table 2 ijerph-18-07028-t002:** Parameters used to calculate cases avoided, base scenario settings (all screenings).

Parameter	Value in Base Scenario
Tests performed	125,865
Contact-tracers	200
Cost per test	€58.32 *
Cost of one contact-tracer per day	€129
Cost of 10 follow-up calls to COVID-19 positives treated at home	€280
Cost of a COVID-19 hospitalization	€6050
Cost of admission to ICU because of COVID-19	€43,400
Cost of permanent sequelae from COVID-19 discounted 3%	€14,754
Positivity rate	1.37%
Average no. of close contacts per COVID-19 case	3
% of close contacts testing positive	24%
% adherence to quarantine	75%
% detected that could infect after detected	80%
Instantaneous effective reproductive number (R_t_)	1.29
Number of iterations	2.58
Hospital admission rate (non-ICU)	3.1%
ICU admission rate	0.2%
Lethality rate	0.6%
Permanent sequelae rate	1.0%
QALY lost due to sequelae discounted at 3%	2.78
QALY lost by mortality at 3%	2.92
Monetary value of a QALY	€25,000

* Rate paid per test by the healthcare service contractor, covering test kit + staff + infrastructure: PCR 75 euros; TAR 15 euros. Price weighted proportionately according to the number of PCR and RAT tests performed.

**Table 3 ijerph-18-07028-t003:** Economic and health consequences of the TTQ strategy in Catalonia.

Results	Quantity	Cost/Unit	Total Cost
Cost of mass testing and contact-tracing			
Tests	125,865	€58.32 *	€7,340,265
Contact-tracers	200.00	€129	€1,032,000
Avoided health consequences			
Total COVID-19 cases	5429		
COVID-19 cases treated at home	5161	€280	€1,445,148
Hospitalizations	168	€6050	€1,018,224
Admission to the ICU	11	€43,400	€471,244
Cases with permanent sequelae	56	€14,754	€827,706
Deaths	33		
Health Improvements (QALY)			
QALY gained by avoided morbidity	156		
QALY earned for avoided mortality	95		
Total monetary costs			€8,372,265
Total monetary savings			€3,762,322
Increase in costs			€4,609,943
Increase in health improvements (total QALY)	251		
Cost per QALY earned			€18,392
Benefit-cost ratio excluding monetized health 0.45			
Benefit-cost ratio including monetized health 1.20			

* Price weighted proportionately according to the number of PCR and RAT tests performed.

**Table 4 ijerph-18-07028-t004:** Summary of results and sensitivity analysis of the cost effectiveness of a TTQ strategy, broken down by test type and wave of epidemic.

Scenario	Differences in Parameters, wrt Base Case	Benefit-Cost Ratio Excluding Health Impacts	Benefit-Cost Ratio (inc. Health Impacts)
Base case		0.45	1.20
PCR	Test cost: €75 Positivity rate: 1.66% Instantaneous effective reproductive number: 1.33	0.46	1.23
RAT	Test cost: €15 Positivity rate: 0.5% Close contacts that give positive: 48% (×2 wrt base case) Instantaneous effective reproductive number: 1.18	0.61	1.63
First Wave	Positivity rate: 1.94% Instantaneous effective reproductive number: 1.39	0.56	1.49
Second Wave	Positivity rate: 1.54% Instantaneous effective reproductive number: 1.02	0.33	0.87

## Data Availability

Not applicable.

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
