# Peer review of "A Cost-Benefit Analysis of the COVID-19 Asymptomatic Mass Testing Strategy in the North Metropolitan Area of Barcelona"

_ijerph, 2021, doi:10.3390/ijerph18137028_

Round 1

Reviewer 1 Report

Review of the paper entitled “A Cost-Benefit Analysis of the COVID-19 Asymptomatic Mass Testing Strategy in the North Metropolitan Area of Barcelona”

The paper examines the economic impact of mass COVID-19 screening of an asymptomatic population in Catalonia, Spain. The results showed that the Benefit-Cost ratio would be 0.45 if the health benefits of avoided COVID-19 cases are not added in the benefit stream and would be 1.20 if these benefits are monetized and included in the benefit. The authors used more than 90% of their parameters from published sources.

Overall, the paper is well-written and timely. I suggest publication. I’ve only one comment. The analysis is at the society level. Therefore, the cost should include the time cost of individuals who took the test.

Author Response

REVIEWER 1

The paper examines the economic impact of mass COVID-19 screening of an asymptomatic population in Catalonia, Spain. The results showed that the Benefit-Cost ratio would be 0.45 if the health benefits of avoided COVID-19 cases are not added in the benefit stream and would be 1.20 if these benefits are monetized and included in the benefit. The authors used more than 90% of their parameters from published sources.

Overall, the paper is well-written and timely. I suggest publication. I’ve only one comment. The analysis is at the society level. Therefore, the cost should include the time cost of individuals who took the test.

- Dear Reviewer 1,
Thank you very much for your contributions. In relation to your comment, the aspect regarding the time cost of individuals was rejected due to the debatable opportunity cost of the citizens at the time of making the tests. In addition, this parameter is not incorporated in the reference model (Vallejo et al, 2021). Thus, we have decided to incorporate this issue into the discussion.
Thank you so much for this suggestion, on behalf of all the co-authors.

Reviewer 2 Report

The authors approached a cost benefit analysis related to the mass testing strategy of asymptomatic in the context of Covid-19 pandemics, in the metropolitan area of Barcelona. The article is well structured, representative for the strategic actions regarding the prevention of spreading the virus during a period of sanitary crisis.

The analysis is relevant, well substantiated, the article complies with rules of drafting a scientific article. The only detected vulnerability related to the methodology is the histogram distribution of the opportunity of implementing the strategy, considering the variation of the infection rate, mainly the dynamic estimation of the cost – benefit rate, including the cost of the sanitation monitoring. It is recommended that this aspect is treated at least in relation to the limitations of the study, if modelling within the methodology is not possible.

Author Response

REVIEWER 2

The authors approached a cost benefit analysis related to the mass testing strategy of asymptomatic in the context of Covid-19 pandemics, in the metropolitan area of Barcelona. The article is well structured, representative for the strategic actions regarding the prevention of spreading the virus during a period of sanitary crisis.

The analysis is relevant, well substantiated, the article complies with rules of drafting a scientific article. The only detected vulnerability related to the methodology is the histogram distribution of the opportunity of implementing the strategy, considering the variation of the infection rate, mainly the dynamic estimation of the cost – benefit rate, including the cost of the sanitation monitoring. It is recommended that this aspect is treated at least in relation to the limitations of the study, if modelling within the methodology is not possible.

- Dear Reviewer 2,
Thank you very much for your contributions. In relation to your comment, we want to emphasize that the methodology used is identical to the one developed in Vallejo et al. (2021). In this sense, we know that it is a simple model but our intention was to apply a methodology previously validated in the literature. In the discussion presented, two paragraphs are devoted to listing the shortcomings of this model. We hope, in this way, to have responded to your concern.
Thank you so much for this suggestion, on behalf of all the co-authors.

Reviewer 3 Report

Thank you for giving me the opportunity to review your work. The manuscript is interesting but I have some few comments below that need to be addressed.

Abstract:

  1. Abstract is too long. Reasonably shorten the abstract. Focus on the motivation for the study, method, results and implication

Introduction

  1. The authors should provide a significant benefits of the study.
  2. I will be glad if the authors can provide the innovation of the study.

Methods

  1. The methods of the study should elaborate on the procedures used in collecting the data of the study.
  2. The researchers should indicate how the study variables were measured

Conclusion

  1. The study should include practical and theoretical implications of the study.

Author Response

REVIEWER 3

Thank you for giving me the opportunity to review your work. The manuscript is interesting but I have some few comments below that need to be addressed.

- Dear Reviewer 3,
From all the co-authors, thank you very much for your comments, which we proceed to respond below.

Abstract: Abstract is too long. Reasonably shorten the abstract. Focus on the motivation for the study, method, results and implication

- The abstract has been reduced. Thank you so much for your comment

Introduction: The authors should provide a significant benefits of the study. I will be glad if the authors can provide the innovation of the study. 

- The paper states that "There is no evidence, however, regarding the economic impact of mass screening in asymptomatic populations. In this context, the aim of this study is to undertake a cost-benefit analysis of the mass COVID-19 screenings in the asymptomatic general population". The article has motivated that a lot of expenditure has been executed based on population screening programs, and instead no evidence has been calculated regarding the economic impact of doing so in asymptomatic population.

Methods: The methods of the study should elaborate on the procedures used in collecting the data of the study.
The researchers should indicate how the study variables were measured.

- Section 2.2 "Study Parameters" lists the origin of the parameters used (first paragraph: observational, based on administrative data; second paragraph: assumptions, some based on other studies).

Conclusion: The study should include practical and theoretical implications of the study.

- The "Conclusions" section, with an eminently practical will, suggests that COVID-19 screening programs need to be modelled on proven successful testing interventions and the results of this study show that an efficient implementation of mass screening in asymptomatic populations can generate high social returns, with the inclusion of avoided impacts on health being the factor that critically determines its desirability.

As a take away, it is stressed out that maximizing the value of dedicated re-sources depends on screening strategies being accompanied by contact-tracing and specific in their focus, targeting, for example, high-risk subpopulations with the high-est rate of expected positives.

Reviewer 4 Report

The authors should clarify:

1) how they calculated the cost-benefit ratio;

2) how QALY was monetized (e.g., willingness-to-pay?);

3) the reasons why they used the cost-benefit approach instead of the most common cost per QALY (i.e., cost-utility analysis);

4) in relation to point 3), using both approaches (CBA and CUA) could provide an interesting methodological comparison.

Author Response

REVIEWER 4
The authors should clarify:

- Dear Reviewer 4,
From all the co-authors, thank you very much for your comments, which we proceed to respond below.

1) how they calculated the cost-benefit ratio;

- The cost-benefit ratio is calculated by dividing the sum of the total benefits by the sum of the total costs. This is an element that has not been made explicit in the text, as it is taken for granted that the reader knows what a ratio is.

2) how QALY was monetized (e.g., willingness-to-pay?);

- This issue has been specified in the text. Thank you for your comment.

3) the reasons why they used the cost-benefit approach instead of the most common cost per QALY (i.e., cost-utility analysis);

- This is due to two reasons. First, the Cost-Utility Analysis (CUA) is an economic evaluation that measures QALY as the only impact variable. Cost-Benefit Analysis, instead, is able to incorporate the monetization of this and other impacts (in this analysis, the main one is avoided health consequences). Methodologically, CBA is superior to CUA, as is able to take into account more elements.

Second, CUA only makes sense if two alternative interventions are analyzed. In this case there is a single object of analysis (mass screening strategies), so the only possible methodology is CBA.

4) in relation to point 3), using both approaches (CBA and CUA) could provide an interesting methodological comparison.

- Answered above.

Round 2

Reviewer 4 Report

Thanks for replying to my previous comments. However, it is not sufficient to say "monetized according to the reference study", you should explain the methodology also in this study. 

Author Response

The sentene has been rewritten again, hoping it now fits the expetations of the reviewer.
